

# Expression of the neurotrophic tyrosine kinase receptors, *ntrk1* and *ntrk2a*, precedes expression of other *ntrk* genes in embryonic zebrafish

Katie Hahn, Paul Manuel and Cortney Bouldin

Department of Biology, Appalachian State University, Boone, NC, USA

## ABSTRACT

**Background:** The neurotrophic tyrosine kinase receptor (*Ntrk*) gene family plays a critical role in the survival of somatosensory neurons. Most vertebrates have three *Ntrk* genes each of which encode a Trk receptor: TrkA, TrkB, or TrkC. The function of the Trk receptors is modulated by the p75 neurotrophin receptors (NTRs). Five *ntrk* genes and one p75 NTR gene (*ngfrb*) have been discovered in zebrafish. To date, the expression of these genes in the initial stages of neuron specification have not been investigated.

**Purpose:** The present work used whole mount in situ hybridization to analyze expression of the five *ntrk* genes and *ngfrb* in zebrafish at a timepoint when the first sensory neurons of the zebrafish body are being established (16.5 hpf). Because expression of multiple genes were not found at this time point, we also checked expression at 24 hpf to ensure the functionality of our six probes.

**Results:** At 16.5 hpf, we found tissue specific expression of *ntrk1* in cranial ganglia, and tissue specific expression of *ntrk2a* in cranial ganglia and in the spinal cord. Other genes analyzed at 16.5 hpf were either diffuse or not detected. At 24 hpf, we found expression of both *ntrk1* and *ntrk2a* in the spinal cord as well as in multiple cranial ganglia, and we identified *ngfrb* expression in cranial ganglia at 24 hpf. *ntrk2b*, *ntrk3a* and *ntrk3b* were detected in the developing brain at 24 hpf.

**Conclusion:** These data are the first to demonstrate that *ntrk1* and *ntrk2a* are the initial neurotrophic tyrosine kinase receptors expressed in sensory neurons during the development of the zebrafish body, and the first to establish expression patterns of *ngfrb* during early zebrafish development. Our data indicate co-expression of *ntrk1*, *ntrk2a* and *ngfrb*, and we speculate that these overlapping patterns indicate relatedness of function.

## INTRODUCTION

Diverse somatosensory receptors are responsible for sensing an organism's environment as well as protecting it from harmful stimuli. Despite the diversity, the variety of receptors can be divided by function into three main groups: mechanoreceptors sense mechanical

Corresponding author
Cortney Bouldin,
bouldinc@appstate.edu

stimuli to provide information about touch, pressure, and vibration; proprioceptors provide information about the positioning of the body in space, and nociceptors sense harmful stimuli to initiate the sensation of pain (*Purves et al., 2001a*). The neurotrophic receptor tyrosine kinase (*Ntrk*) genes, which encode the Trk receptors, are essential for survival, proliferation and continued differentiation of sensory neurons (*Huang & Reichardt, 2001*).

Most vertebrates (including mammals) have three Trk receptors (TrkA, TrkB, and TrkC), which are activated by neurotrophin ligands (*Hallböök, 1999*; *Benito-Gutiérrez, Garcia-Fernàndez & Comella, 2006*). Nerve growth factor (NGF) binds to TrkA; brain-derived neurotrophic factor (BDNF) and neurotrophin-4 (NT-4) bind to TrkB; and neurotrophin-3 (NT-3) binds to TrkC (*Bibel & Barde, 2000*; *Huang & Reichardt, 2003*). The functions of the Trk receptors were determined through experiments that used targeted gene deletion in mice to disrupt the receptors, and in some cases, the respective ligands. Knockout of TrkA and NGF results in a loss of neurons typically noted for their role in nociception (*Smeyne et al., 1994*; *Crowley et al., 1994*), knockout of TrkB and BDNF reduced sensory neurons involved in mechanoreception (*Klein et al., 1993*; *Ernfors, Lee & Jaenisch, 1994*; *Jones et al., 1994*), and knockout of TrkC reduced neurons that play a role in proprioception (*Klein et al., 1994*), leading to a modular model of function, where expression of each Trk receptor was responsible for a specific subset of sensory neurons (*Klein, 1994*).

More recently, the modular model of Trk gene function has been complicated by the establishment of crosstalk among the neurotrophin signaling pathways and other signaling pathways; the identification of multiple Trk splice isoforms; and the discovery of p75 NTR, which modulates the function of each Trk (*Stucky & Koltzenburg, 1997*; *Chao, 2003*). p75 NTR is capable of binding with low affinity to each of the neurotrophins that interact with Trk receptors (*Schecterson & Bothwell, 2010*). In the presence of TrkA, p75 NTR will increase the affinity of the receptor for NGF (*Hempstead et al., 1991*; *Bibel, Hoppe & Barde, 1999*), which results in a prosurvival effect. In other contexts, uncleaved proneurotrophins can bind to p75 NTR and induce death independently of a Trk receptor (*Lee et al., 2001*; *Teng et al., 2005*). Much work has gone into investigating the effects of p75 NTR on survival in cell culture. Interactions between Ntrks and p75 NTR in vivo have been slow to materialize (*Bothwell, 2016*) but are increasingly being better understood (*Tanaka et al., 2016*; *Chen et al., 2017*). Zebrafish is proving to be a useful model system for accelerating discovery of the interplay between Trk receptors and p75 NTR in vivo (*Anand & Mondal, 2020*).

Using mammalian model systems, the *Ntrk* and *p75 NTR* genes have been established as vital for neurogenesis, as their expression is necessary for sensory neuron survival. Unlike mammals, zebrafish have undergone a whole-genome duplication event that happened more than 300 million years ago in the teleost lineage (*Amores et al., 1998*). As a result, zebrafish have five *ntrk* genes (*ntrk1*, *ntrk2a*, *ntrk2b*, *ntrk3a*, and *ntrk3b*), and so five Trk receptors (TrkA, TrkB1, TrkB2, TrkC1, and TrkC2; *Hallböök, 1999*). We elected to study the expression of the five *ntrk* genes during zebrafish embryogenesis by looking at

two time points in early development when the sensory neurons are found in the zebrafish embryo (*Metcalfe & Westerfield, 1990*; *Kimmel et al., 1995*). Near the time when sensory neurons can first be detected, we found previously undescribed expression of *ntrk1* and *ntrk2a*. Later in embryogenesis, when sensory motor reflexes are beginning, we found expression of *ntrk1*, *ntrk2a* and the p75 NTR homolog, *ngfrb*, in cranial ganglia and the spinal cord. We also confirmed that *ntrk2b*, *ntrk3a* and *ntrk3b* were expressed in the developing brain.

## MATERIALS AND METHODS

### Fish care

Fish were kept between 26 °C and 28 °C in a 14 h light/10 h dark cycle and were cared for according to *The Zebrafish Book: A guide for the laboratory use of zebrafish (Danio rerio)* (*Westerfield, 2000*). Water temperature was kept between 26 °C and 28 °C, conductivity was maintained between 450 and 650 microSiemens and pH was kept above 6.8. Fish were fed dry food daily at 9 am and live brine shrimp in the afternoon. All zebrafish use was approved by the Appalachian State University Institutional Animal Care and Use Committee (Protocols 17-13 and 17-15).

### Embryo collection

Male and female adult zebrafish were separated in breeding tanks in a 2:3 ratio after the afternoon feeding. Timed matings were used to collect embryos. Fertilized, live embryos were then moved to embryo media (15 mM NaCl, 0.8 mM KCl, 1.3 mM $CaCl_2 \cdot 2H_2O$, 0.1 mM $KH_2PO_4$, 0.05 mM $Na_2HPO_4$, 2 mM $MgSO_4 \cdot 7H_2O$, 0.07 mM $NaHCO_3$) and moved to a 28 °C incubator.

### Embryo staging and fixation

Embryos were staged according to *Stages of embryonic development of the zebrafish* (*Kimmel et al., 1995*). Embryos at 16.5 hpf (the 15 somite stage) were distinguished by the number of somites present, as well as by yolk extension length. By Prim-5 stage or 24 hpf, embryos have about 30 somites, faint melanin pigment, and a distinct angle between the head and trunk. Embryos were fixed at the appropriate stage using a 4% paraformaldehyde solution overnight at 4 °C before dehydration into methanol and long-term storage at −20 °C.

### cDNA isolation and cloning

The *ntrk1* clone was purchased from Dharmacon (clone ID: 9038008). For all others, cDNA was isolated from zebrafish embryos using Superscript III First-Strand Kit (Invitrogen). *ntrk2a, ntrk2b, ntrk3a*, and *ntrk3b* were cloned from cDNA using Gateway pENTR/D-TOPO Cloning Kit (Invitrogen, Carlsbad, CA, USA). All forward primers contained the sequence CACC for directional cloning with the kit. The sequence of the primers used can be found in Table 1, and a diagram of each gene and probe can be found in Fig. S1.

**Table 1 *ntrk* cloning primers.**

| Gene name | Forward primer | Reverse primer |
| --- | --- | --- |
| *ntrk2a* | CACCATGAGCTTCGGCATGAC | TTATCCCAGGATGTCCAGA |
| *ntrk2b* | CACCATGACCGCAGGGGTTC | TTAGCCCAGGATGTCCAG |
| *ntrk3a* | CACCATGGATTTATTCTCCATCCC | CTAGCCCAGGATATCCAG |
| *ntrk3b* | CACCGGACTTTAAGTGCCTGC | AACATTTAAATCCAACAGGTG |

## Probe synthesis

Sense and anti-sense RNA probes were generated for each of the genes of interest. RNA probes were synthesized with 10 µg of plasmid DNA. Plasmid DNA was linearized using restriction digest. Linearized DNA was extracted using a phenol/chloroform extraction. DNA was resuspended and used in a transcription reaction. The transcription reaction used 10x RNA polymerase buffer, 10x DIG labeling mix (Roche, Basel, Switzerland), RNAsin (Invitrogen, Carlsbad, CA, USA), and T7 or T3 polymerase (New England Biolabs, Ipswich, MA, USA). Cleanup of the synthesized probe was performed using precipitation with LiCl. For the non-hydrolyzed probe (*ntrk2a*), the pellet was resuspended in water and hybridization buffer was added. For hydrolyzed probes (*ntrk1, ntrk2b, ntrk3a, ntrk3b* and *ngfrb*), the pellet was resuspended in water and then hydrolyzed with sodium bicarbonate and sodium carbonate. Hydrolysis occurred by adding 5 µl 0.4 M $NaHCO_3$ and 5 µl 0.6 M $Na_2CO_3$ and incubating at 60 °C for *t* minutes where *t* = (starting kb − desired kb)/(0.11 * starting kb * desired kb) with 0.35 kb used as desired size. Hydrolyzed probes were cleaned up with a NucleoSpin RNA Clean-up kit (Macherey-Nagel, Düren, Germany).

## In situ hybridization and image collection

Single-probe whole-mount in situ hybridizations were performed as described by *Thisse & Thisse (2008)*. For imaging, all embryos were moved stepwise to a solution of 75% glycerol and 25% PBS and left overnight. The following day embryos were mounted on a 1.0 mm thick Gold Seal slide and covered with a Corning 22 × 22 mm cover slip. All embryos were imaged on an Olympus IX81 at 4× or 10× magnification and processed with Olympus cellSens software.

# RESULTS

## *ntrk1, ntrk2a* and *ngfrb* RNA expression

Zebrafish *ntrk* gene expression was examined at two points in early development, 16.5 hpf and 24 hpf, using whole mount in situ hybridization. At 16.5 hpf, embryos are undergoing embryonic patterning and both *neurogenin* and *neuroD* can be found in neurogenic placodes (*Metcalfe & Westerfield, 1990*; *Kimmel et al., 1995*; *Andermann, Ungos & Raible, 2002*). By 24 hpf, embryos have completed embryonic patterning and sensory motor reflexes are beginning (*Kimmel et al., 1995*), making these useful time points to analyze gene expression during the early stages of neuron specification.

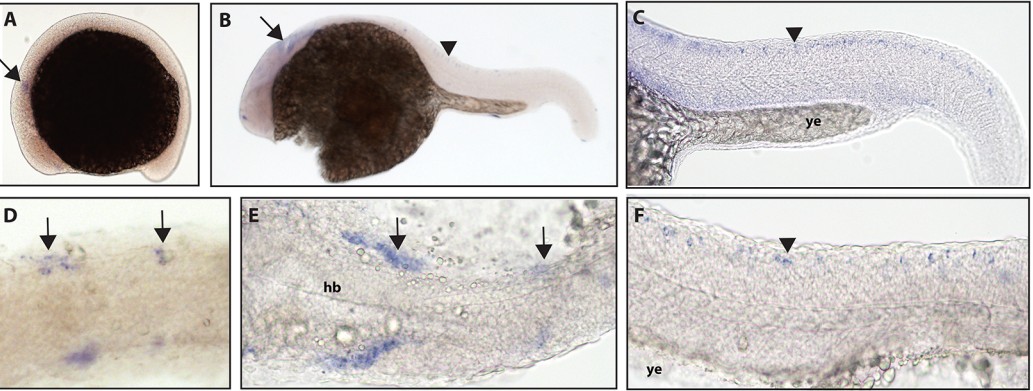

**Figure 1 Early expression of *ntrk1* is seen in the head and spinal cord using whole mount in situ hybridization at 16.5 hpf and 24 hpf.** (A and D) Expression of *ntrk1* is seen at 16.5 hpf is seen in two sets of cranial ganglia (arrow in A and arrows in B). (B, C, E and F) At 24 hpf, expression is seen in two sets of cranial ganglia (arrows) as well as in the RB neurons (arrowheads). (A, B, C and F) Lateral view. (D and E) Dorsal view. All embryos are oriented with the head to the left. hb, marks the hindbrain, and ye, marks the yolk extension. Data are representative of three rounds of in situ hybridization with least 25 embryos per round.               

In zebrafish, only one *ntrk1* gene is present (*Martin et al., 1995*). Expression of the *Ntrk1* gene in mice has been implicated in the survival of nociceptors (*Smeyne et al., 1994*; *Crowley et al., 1994*), the sensory neurons responsible for detecting potentially harmful stimuli (*Purves et al., 2001b*). Zebrafish *ntrk1* expression has not been previously investigated at 16.5 hpf, but has been seen in cranial ganglia, trigeminal ganglia, and Rohon–Beard (RB) neurons at 24 hpf (*Nittoli et al., 2018*). Using a full-length hydrolyzed *ntrk1* probe, we found expression at 16.5 hpf in two sets of cranial ganglia and at 24 hpf in two sets of cranial ganglia as well as expression consistent with the RB neurons of the spinal cord (Fig. 1).

Zebrafish have two paralogs of the *Ntrk2* gene, *ntrk2a* and *ntrk2b* (*Martin et al., 1995*). Because *ntrk2a* is most closely related to *Ntrk2* in other species, these genes have been presumed to functional similarly. Previous studies have identified expression of *ntrk2a* in the trigeminal ganglia and RB neurons as early as 24 hpf (*Martin et al., 1995*; *Nittoli et al., 2018*). Using whole mount in situ hybridization with a partial length probe, which is designed to hybridize to the extracellular domain and the kinase domain of *ntrk2a*, we found *ntrk2a* expression consistent with expression in the trigeminal ganglia and RB neurons at 16.5 hpf. At 24 hpf, we were also able to confirm expression in cranial ganglia and RB neurons (Fig. 2).

*ngfrb/p75 NTR* is a tumor necrosis factor receptor which binds with low-affinity to each of the neurotrophins (Ngf, Bdnf, NT-4/5 and NT-3; *McKay et al., 1996*). Using a full-length hydrolyzed probe, no specific *p75* NTR expression was detected at 16.5 hpf with diffuse staining observed throughout the embryo (see Fig. S2 for a comparison of sense and anti-sense probes), and by 24 hpf, expression was found in two sets of cranial ganglia (Fig. 3).

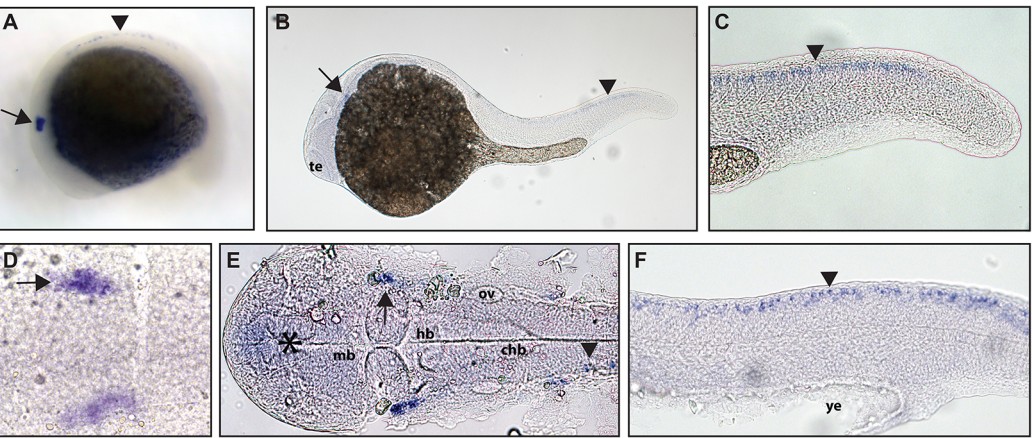

**Figure 2 Early expression of *ntrk2a* is seen in the head and spinal cord using whole mount in situ hybridization at 16.5 hpf and 24 hpf.** Expression of *ntrk2a* is seen in the cranial ganglia (arrows) as well as in the RB neurons of the spinal cord (arrowheads) at (A and D) 16.5 hpf and (B, C, E and F) 24 hpf. The asterisk in E marks weak expression in the diencephalon that matches with previously published accounts. (A, B, C and F) Lateral view. (D and E) Dorsal view. All embryos are oriented with the head to the left. te, marks the telencephalon, mb, marks the midbrain, hb, marks the hindbrain, ov, marks the otic vesicle, chb, marks the caudal hindbrain and ye, marks the yolk extension. Data are representative of two rounds of in situ hybridization with at least 25 embryos per round.

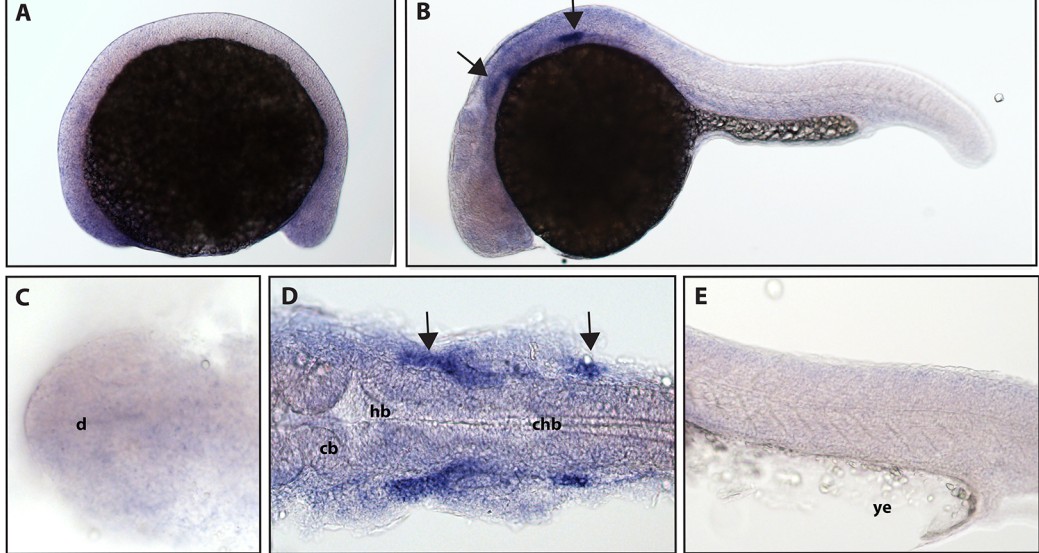

**Figure 3 Early expression of p75 NTR (*ngfrb*) is seen in two sets of cranial ganglia using whole mount in situ hybridization at 24 hpf.** (A and C) Expression of ngfrb was not detected at 16.5 hpf and the staining that can be seen was non-specific (Fig. S2). (B, D and E) At 24 hpf, expression of ngfrb was seen in two sets of cranial ganglia (arrows). (A, B and E) Lateral view. (C and D) Dorsal view. All embryos are oriented with the head to the left. d, marks the diencephalon, cb, marks the cerebellum, hb, marks the hindbrain, chb, marks the caudal hindbrain and ye, marks the yolk extension. Data are representative of the results from two rounds of in situ hybridization with at least 25 embryos per round.

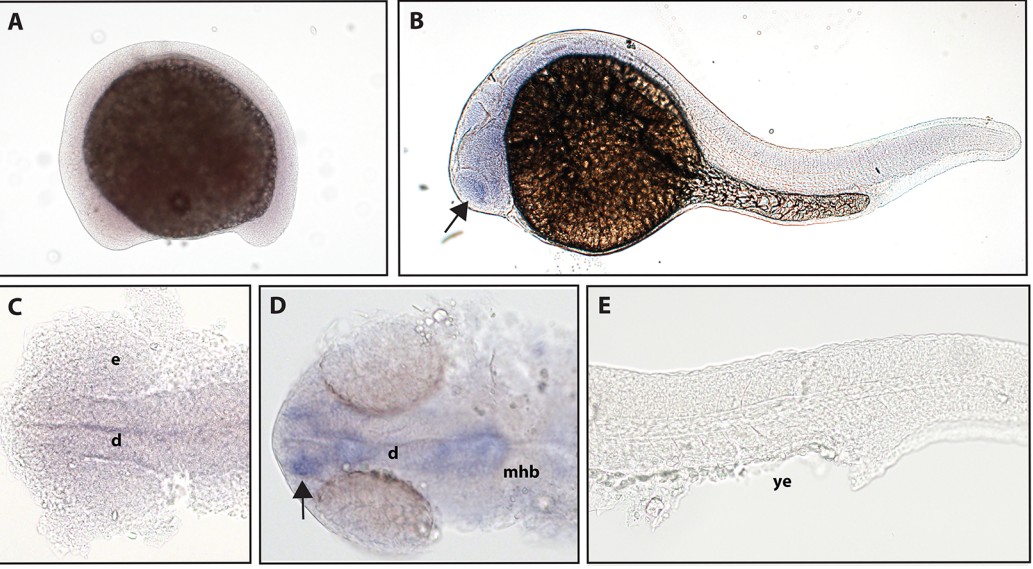

**Figure 4 Early expression of *ntrk2b* is seen in the head using whole-mount in situ hybridization at 24 hpf.** (A and C) Expression of *ntrk2b* was not detectable at 16.5 hpf. (B, D and E) At 24 hpf, expression of *ntrk2b* is seen in the telencephalon (arrows). (A, B and E) Lateral view. (C and D) Dorsal view. e, marks the eye, d, marks the diencephalon, mhb, marks the midbrain-hindbrain boundary, and ye, marks the yolk extension. Data are representative of three rounds of in situ hybridization with at least 25 embryos per round.                                           

## *ntrk2b*, *ntrk3a* and *ntrk3b* RNA expression

The function of *ntrk2b* is unknown, and the expression is distinct from that of *ntrk2a* (*Nittoli et al., 2018*). Other studies using partial sequence probes suggest that expression is found in the telencephalon, thalamus, hypothalamus, tegmentum, hindbrain, cranial nerves, and RB neurons (*Nittoli et al., 2018*). Using a full-length hydrolyzed probe, no expression of *ntrk2b* was seen at 16.5 hpf, and expression at 24 hpf was confirmed in the telencephalon (Fig. 4).

Zebrafish have two paralogs of the *Ntrk3* gene, *ntrk3a* and *ntrk3b* (*Martin et al., 1995*). Because of sequence similarity, *ntrk3a* has been presumed to be functionally similar to *Ntrk3* in other species. Previous studies have determined expression of *ntrk3a* in the telencephalon, pineal gland, hypothalamus, cranial ganglia, and RB neurons at 24 hpf (*Nittoli et al., 2018*). Using a full-length hydrolyzed probe, we found no expression of *ntrk3a* at 16.5 hpf, and at 24 hpf, we were able to confirm expression in the telencephalon and expression consistent with RB neurons (Fig. 5).

The function of *ntrk3b* is unknown, and previous studies have identified some overlap in expression with *ntrk3a*; other studies identified expression in the telencephalon, thalamus, tegmentum, and otic vesicle at 24 hpf (*Nittoli et al., 2018*). Using a hydrolyzed partial length probe, we found no expression of *ntrk3b* at 16.5 hpf, confirmed expression in the telencephalon and otic vesicle, and at 24 hpf, we identified expression in the midbrain and hindbrain (Fig. 6).

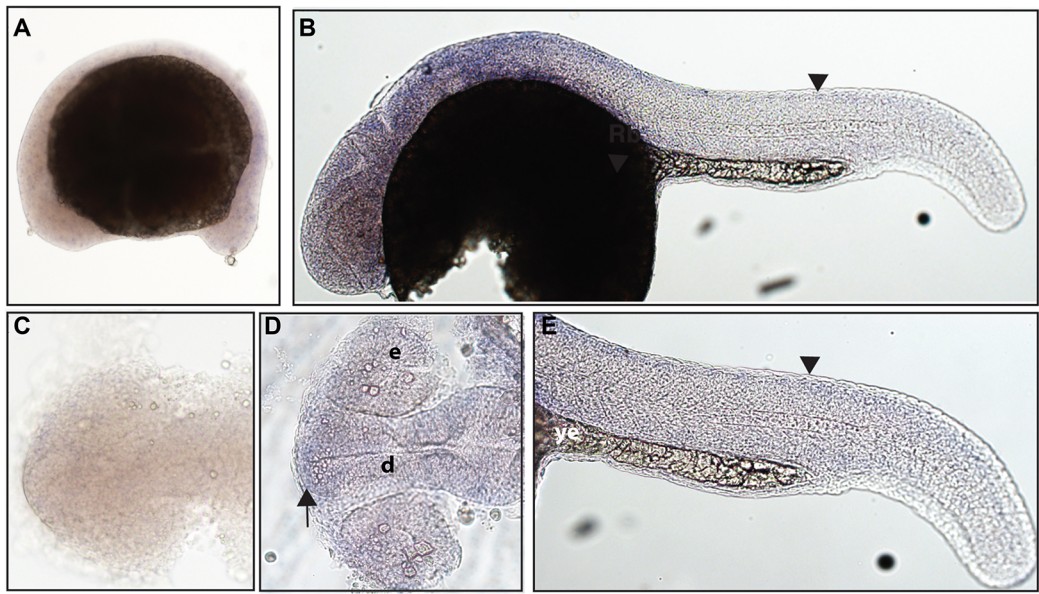

**Figure 5 Early expression of *ntrk3a* is seen in the head and spinal cord using whole-mount in situ hybridization at 24 hpf.** (A and C) Expression of *ntrk3a* was not detected at 16.5 hpf. (B, D and E) At 24 hpf, expression was seen in the telencephalon (arrow) and spinal cord (arrowheads). (A, B and E) Lateral view. (C and D) Dorsal view. All embryos are oriented with the head to the left. e, marks the eye, d, marks the diencephalon, and ye, marks the yolk extension. This data is representative of the results from three rounds of in situ hybridization with at least 25 embryos per round.

## DISCUSSION

When compared to other vertebrates, the teleost lineage has undergone an additional whole-genome duplication event (*Taylor et al., 2001*), and the single *ntrk1* gene found in zebrafish may be due to gene loss after the duplication (*Heinrich & Lum, 2000*). *Ntrk1* in mice has been shown to be important for the survival of nociceptors (*Reichardt & Isabel, 1998*). While the expression of zebrafish *ntrk1* has been characterized at 24 hpf, expression at 16.5 hpf was not prior to this study, making the current work valuable for understanding *ntrk1* expression at a timepoint when sensory neurons are first identified in the zebrafish body. Gene expression at 16.5 hpf was seen in two sets of cranial ganglia, with one larger set more anterior and a smaller set more posterior. Expression at 16.5 hpf, which resembles cranial ganglia expression, indicates that *ntrk1* expression is present early in neurogenic placode formation. Expression of *ntrk1* at 24 hpf was seen in two sets of cranial ganglia as well as in the spinal cord in a manner consistent with expression in the RB neurons.

*Ntrk2* and *Bdnf* in mice have been shown to be critical for mechanoreceptor function (*Klein et al., 1993*; *Ernfors, Lee & Jaenisch, 1994*; *Jones et al., 1994*), with *ntrk2a* being most closely related to *Ntrk2* in other species (*Nittoli et al., 2018*). *ntrk2a* expression was seen in cranial ganglia, which are in a location consistent with trigeminal ganglia, and in the spinal cord in a pattern which resembles RB neurons at both 16.5 hpf and 24 hpf. Expression of *ntrk2a* at 16.5 hpf is unique as it is the only *ntrk* gene that we saw expressed

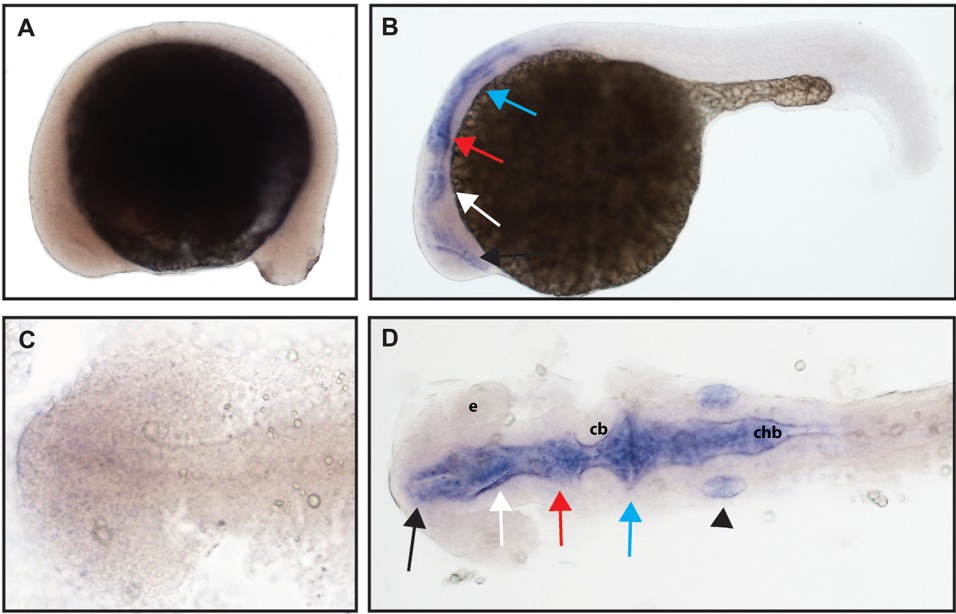

**Figure 6 Early expression of *ntrk3b* is seen throughout the head at 24 hpf using whole mount in situ hybridization.** (A and C) At 16.5 hpf, no expression was detected. (B and D) Expression was seen in the telencephalon (black arrows), the diencephalon (white arrows), the midbrain (red arrows), the hindbrain (blue arrows), and otic vesicle (black arrowhead). (A and B) Lateral view. (C and D) Dorsal view. e, marks the eye, cb, marks the cerebellum, and chb, marks the caudal hindbrain. All embryos are oriented with the head to the left. This data is representative of the results of three rounds of in situ hybridization with at least 25 embryos per round.          

**Table 2 *ntrk* expression overlaps with known expression patterns of the Trk ligands.**

| Gene name | Corroborated gene expression locations at 24 hpf | Genes expressed in similar tissues | Found at 16.5 hpf? |
|---|---|---|---|
| *ntrk1* | Cranial ganglia[a,b] | *ntrk2a*[a,b], *ngfrb*[a] | Yes[a] |
| | Spinal cord[a,b] | *ntrk2a*[a,b], *ntrk3a*[a,b], *bdnf*[c] | No[a] |
| *ntrk2a* | Cranial ganglia[a,b] | *ntrk1*[a,b], *ngfrb*[a] | Yes[a] |
| | Spinal cord[a,b] | *ntrk1*[a,b], *ntrk3a*[a,b], *bdnf*[c] | Yes[a] |
| *ntrk2b* | Forebrain[a,b] | *ntrk3a*[a,b], *ntrk3b*[a,b], *ngf*[b], *bdnf*[c] | No[a] |
| *ntrk3a* | Forebrain[a,b] | *ntrk2b*[a,b], *ntrk3b*[a,b], *ngf*[b], *bdnf*[c] | No[a] |
| | Spinal cord[a,b] | *ntrk2a*[a,b], *ntrk3a*[a,b], *bdnf*[c] | No[a] |
| *ntrk3b* | Forebrain[a,b] | *ntrk2b*[a,b], *ntrk3b*[a,b], *ngf*[b], *bdnf*[c] | No[a] |
| | Otic vesicle[a,b] | *ngf*[b], ntf3[b], *ntf6/7*[b], *bdnf*[c] | No[a] |

**Notes:**
[a] The present work.
[b] *Nittoli et al., 2018*.
[c] *De Felice et al., 2014*.

in the spinal cord at this time point and early expression suggests an important role for *ntrk2a* in development of the early peripheral nervous system. Expression of *ntrk2a* in the trigeminal overlaps with expression of NT-4/5, a TrkB ligand (Table 2; *Nittoli et al., 2018*).

*ntrk2b* expression was detected in a completely different location from *ntrk2a*, suggesting the possibility of functional differences. No expression of *ntrk2b* was detected at 16.5 hpf and expression at 24 hpf was seen in the telencephalon. Although *ntrk2a* is most closely related to *Ntrk2*, the TrkB ligand, BDNF, overlaps with *ntrk2b* and not *ntrk2a* at 24 hpf (Table 2; *De Felice et al., 2014*). Interestingly, coexpression of *bdnf* and *ntrk2b* persists in many tissues into adulthood (*Sahu et al., 2019*).

*ntrk3a* is the gene most closely related to *Ntrk3* in other species (*Nittoli et al., 2018*). In mice, *Ntrk3* has been shown to be involved in the survival of proprioceptors (*Klein et al., 1994*), but the functionality of *ntrk3a* and *ntrk3b* in zebrafish is unknown. Neither *ntrk3a* nor *ntrk3b* expression was detected at 16.5 hpf. At 24 hpf, diffuse staining can be seen in *ntrk3a* embryos throughout the head, with the most specific staining appearing to be located in the telencephalon. BDNF expression overlaps with *ntrk3a* expression in the telencephalon (Table 2; *De Felice et al., 2014*; *Nittoli et al., 2018*). Expression of *ntrk3a* can also be seen in the spinal cord at 24 hpf. Compared to *ntrk1* and *ntrk2a*, *ntrk3a* expression appears more diffuse throughout the spinal cord instead of distinct puncta. Expression of *ntrk3b* was found in the telencephalon, midbrain, hindbrain, and otic vesicles at 24 hpf. Expression in the telencephalon overlaps with known expression of the TrkA ligand NGF and the TrkB ligand BDNF, as well as with NGF and NT-3 in the otic vesicle (Table 2; *Nittoli et al., 2018*).

p75 NTR is a member of the tumor necrosis factor receptor superfamily and is mainly expressed early in development (*Dechant & Barde, 2002*). p75 NTR can interact with all of the *ntrk* genes and can bind with low affinity to NGF, BDNF, and NT-3, the Trk ligands (*Schecterson & Bothwell, 2010*). p75 NTR complicates the idea that each Trk receptors have a modular model of function, with one ligand binding to a Trk receptor to promote survival of a specific subset of sensory neuron, because p75 NTR can promote survival of neurons as well as neuronal death (*Meeker & Williams, 2015*). While p75 NTR complicates the previously understood model, knowing when and where it is expressed will help with our understanding of the *ntrk* genes, including their functionality during early development.

NCBI databases indicate that zebrafish have two p75 NTR homologs, *ngfra* and *ngfrb*. We elected to study expression of *ngfrb* because at the time when we conceived this study *ngfrb* had a reference sequence accession number (NP_001185589.1), while *ngfra* was merely predicted (XP_003198133.2). No expression of *ngfrb* was detected at the 16.5 hpf time point, while expression at 24 hpf can be seen in two domains of staining in the cranial ganglia. Interestingly, expression of *ngfrb* appears similar to the expression pattern determined for *ntrk1* (Table 2). TrkA has been shown to cause a prosurvival effect for p75 (*Hempstead, 2002*). An overlapping expression pattern between *ngfrb* and *ntrk1* would make sense as expression of both TrkA and p75 receptors in a similar location would promote neuron survival.

While previous studies have used partial gene sequences for making probes, a majority of our data was collected using full-length probes that were hydrolyzed to 300–400 bp fragments (Fig. S1). We reasoned that this would be an advantageous approach with the Trk family that has been demonstrated to express multiple splice isoforms

(*Barker et al., 1993*; *McGregor et al., 1994*; *Eide et al., 1996*) because a hydrolyzed full-length probe should not favor any specific gene region or splice isoform. From this we assert that while we identified fewer regions of gene expression than a previous study (*Nittoli et al., 2018*), we have very high confidence in the locations of gene expression that we have identified.

Some studies have suggested that each of the *ntrk* genes are functionally distinct in zebrafish due to differential expression patterns (*Martin et al., 1995*; *Nittoli et al., 2018*), while other studies have suggested that *ntrk1, ntrk2a, and ntrk3a* are all functionally similar due to overlapping expression with important nociceptive ion channels (*Gau et al., 2017*). Our work suggests that there are two groupings of genes with one grouping—*ntrk1*, *ntrk2a* and *ngfrb*—involved in development of the peripheral nervous system and another grouping—*ntrk2b*, *ntrk3a* and *ntrk3b*—involved in the development of the central nervous system.

While we have determined the locations of gene expression at two critical timepoints during development, these efforts do not allow us to definitively determine functionality. Similar to the experiments in mice used to determine functionality, the zebrafish *ntrk* genes could be knocked out. For example, *ntrk2b* has been knocked out and shown to produce anxiety through an effect on dopaminergic and serotonergic neuronal populations (*Sahu et al., 2019*). Our data indicate that *ntrk1* and *ntrk2a*, the two *ntrk* genes that we found to be initially expressed in sensory neurons, would be interesting to knockout and analyze for functions. CRISPR is an accessible tool that would allow these knockouts to be done relatively easily and could provide information about the functionality of these genes (*Hwang et al., 2013*; *Tennant, Zerucha & Bouldin, 2019*). From our data, we hypothesize that both *ntrk1*, because of the similarity of the protein to *Ntrk1*/TrkA, and *ntrk2a*, because of its expression in cranial ganglia, would play a critical role in nociception. Behavioral analysis using zebrafish have been developed (*Curtright et al., 2015*), which could be performed on crispants to test this hypothesis.

## CONCLUSIONS

In our study, we looked at *ntrk* expression in zebrafish at two early time points in development, including 16.5 hpf, which is earlier than seen in any other published account of expression in zebrafish (*Martin et al., 1995*; *Martin, Sandell & Heinrich, 1998*; *Gau et al., 2017*; *Nittoli et al., 2018*). In addition, we used hydrolyzed probes to provide a method of determining expression that would not be limited to a specific splice isoform or homologous region. Through this study, we have investigated embryonic expression of the *ntrk* genes and one *p75 NTR* (*ngfrb*) in zebrafish. We have identified expression of all five *ntrk* genes and *ngfrb* prior to the larval stages, which strengthens the developmental role of these genes in establishing connections between neurons and target cells, and we have established *ntrk1* and *ntrk2a* as the genes expressed in the first sensory neurons of developing zebrafish embryos.

## ACKNOWLEDGEMENTS

The authors would like to thank Andrew Bellemer for helpful discussions and critical reading of the manuscript. Additionally, we would also like to thank Appalachian State

University College of Arts and Sciences Vivarium, Monique Eckerd and Scott Rhyne for assistance in caring for our zebrafish colony, and the College of Arts and Sciences William C. and Ruth Ann Dewel Microscopy Facility and Dr. Guichuan Hou, the director of the Dewel Microscopy Facility.

### Funding

This work was supported by the Department of Biology, the College of Arts and Sciences, and the Office of Student Research at Appalachian State University. The funders had no role in study design, data collection and analysis, decision to publish, or preparation of the manuscript.

### Grant Disclosures

The following grant information was disclosed by the authors:
Appalachian State University.

### Competing Interests

The authors declare that they have no competing interests.

### Author Contributions

- Katie Hahn conceived and designed the experiments, performed the experiments, analyzed the data, prepared figures and/or tables, authored or reviewed drafts of the paper, and approved the final draft.
- Paul Manuel performed the experiments, analyzed the data, prepared figures and/or tables, and approved the final draft.
- Cortney Bouldin conceived and designed the experiments, analyzed the data, authored or reviewed drafts of the paper, and approved the final draft.

### Animal Ethics

The following information was supplied relating to ethical approvals (i.e., approving body and any reference numbers):

All zebrafish use was approved by the Appalachian State University Institutional Animal Care and Use Committee (Protocol 17-13).

### Data Availability

The raw data (unprocessed images of in situ staining results) are available as Figs. 1–6 and in the Supplemental Files.

### Supplemental Information

Supplemental information for this article can be found online at http://dx.doi.org/10.7717/peerj.10479#supplemental-information.

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
