# Peer review of "Expression of the neurotrophic tyrosine kinase receptors, ntrk1 and ntrk2a, precedes expression of other ntrk genes in embryonic zebrafish"

_PeerJ, doi:10.7717/peerj.10479_

## Round 0.1 · original submission · Major Revisions

Thank you for submitting your manuscript to PeerJ. Based on the comments of three reviewers I invite you to resubmit after making major revisions. All three reviewers were positive about your manuscript, but provided a number of recommendations. Please address all reviewer suggestions in your rebuttal letter.

Most significantly, reviewers 1 and 3 have questions about how the in situ images are used to make specific conclusions about tissues expression. Reviewer 2 had concerns about how the in situ probes were validated and how these data are presented, suggested additional experiments to functionally test your conclusions about Trk function and had suggestions for the presentation of in situ data.

I would also suggest comparing your temperospatial gene expression data to two publically available zebrafish gene expression datasets: EBI’s RNA-Seq timeseries and a single cell RNA-Seq dataset on the UCSC Cell Browser that begins with a 24 hpf timepoint. The latter dataset is reported in a recent paper by Farnsworth et al. in Developmental Biology. I have included URLs below in case these are helpful:

https://www.ebi.ac.uk/gxa/experiments/E-ERAD-475
https://cells.ucsc.edu/?ds=zebrafish-dev

·

Basic reporting

1. Overall, this manuscript is nicely written and data provided appear to be of high quality.
Although the paper is generally very well written, lines 213-214 could use some attention with respect to grammar. (Our data…., we found….).

2. Very minor point: The last sentence in the abstract is rather speculative, and might be best identified as such (We speculate rather than we suggest).

3. This study focuses on gene expression rather than genes themselves. Therefore it would be more accurate for the abstract (purpose) to state "Because expression of multiple genes", rather than stating "Because multiple genes were not found"

4. A minor point, but the first results sentence in the abstract uses a different tense than the remainder of that section. (find vs. found). Pointing out in case this is unintentional.

Experimental design

1. The objective of the study was to learn which neurotrophin receptors are expressed during very early neural development in zebrafish. The methodology was exclusively in situ hybridization. Although not necessary, if antibodies are available for any of the studied genes, use could strengthen the conclusions. Based on a quick search within zfin.org, this appears to be the case for ntrk1 and ntrk2.

Validity of the findings

1. The authors define spatiotemporal expression patterns. They conclude that only ntrk1 and ntrk2a are expressed at 16.5 hpf, whereas others become expressed by 24 hpf. They do a good job of relating and comparing their findings to past work.

2. Although this reviewer agrees that the spinal expression probably represents expression in RB neurons, the authors should describe what evidence this is based on. If none, simply changing the language to state that the expression pattern is consistent with that of RB neurons is advised. If the authors feel that certainty is critical for their conclusions, additional markers should be used to demonstrate expression in RB neurons.

3. Overall the work could be strengthened by using alternative markers that support conclusions such as expression in cranial ganglia at 16.5 hpf, or in RB neurons at 24 hpf. Alternatively, softer language when making conclusions could circumvent the lack of direct evidence.

Additional comments

1. This may be beyond the scope of the study, but the results and conclusions would be more interesting and valuable if expression were narrowed to specific cell types or regions. For instance, in the hindbrain (ntrk3b), which rhombomeres or cell types express? This is not necessary, but would again add more value to the study.

2. One thing that could be more clearly stated was whether expression patterns listed on table 2 for BDNF, NGF, and NT-3 extracted from the literature, or were these experiments done by the authors? The legend states “known expression patterns”, but this might be more explicitly stated to avoid confusion (along with citing those studies within the figure or legend if desired by the authors).

3. If the authors are looking to strengthen the discussion, the first paragraph (line 232) could be strengthened by more specifically pointing out at what stages these neurons are undergoing specification, differentiation, axon outgrowth, etc. The statement that earliest specification occurs at 16.5 hours seems unsupported as written.

4. In the abstract, it could be more accurate to use the word functionality rather than specificity. Does testing these at 24 hpf truly demonstrate that they are specific, or just that they can hybridize and produce signal?

5. Overall this is a nice study with quality images and reporting. I hope you will find some comments that you feel improve your paper. -JH Hines

Reviewer 2 ·

Basic reporting

Major concerns:
1. The authors have described literature across species for Trk signaling, and raised the critical point about splice isoforms that their study addresses. However, it seems the authors have missed opportunities to discuss key recent literature in zebrafish that is relevant to their question. Indeed, they say that zebrafish “may provide a potentially useful model”, rather than discussing the literature showing just that – this would be a compelling assertion. For instance, Sahu et al. Sci Rep 2019 describes both morphant and mutant ntrk2b zebrafish and effects on particular neurotransmitter systems (perhaps there other knockouts/knockdowns to discuss?). The authors should also contrast their work with another recent study, Anand & Mondal J Neuro Research 2019.

2. The authors have done a commendable job describing their methodology, but in some places it can be condensed, for instance when describing standard kit directions. Additionally, much of the in situ hybridization protocol seems derived from the Thisse & Thisse 2008 Nature Protocols paper. The Methods section can be condensed if this is the case to highlight any differences the authors made from the “standard” protocol. At minimum, the Thisse & Thisse paper should be cited.

3. The authors state that raw data are included in the form of micrographs. However, it sounds as though they have produced novel in situ probes, which require additional validation to show specificity via sense probes (see below). I think it is most suitable to show sense probe staining in the paper; however, it is at minimum necessary to include in the supplemental data.

Minor comments:
1. I was confused by this sentence in particular in the discussion (lines 246-249): “Ligands and receptors are expressed in different cells so overlapping expression does not necessarily mean overlapping function..” and the surrounding paragraph was confusing. I believe the authors are trying to make a circumspect statement about potential BDNF signaling in the telencephalon via ntrk2b, but the language is confusing. Furthermore, the statement suggests that there cannot be autocrine signaling where there is at least some literature precedent for in neurons (McWilliams et al Cell Rep 2015) and particularly in synapses (Harward et al Nature 2016).

2. There are several uses of “data” as a singular noun, particularly in figure captions. Please revise to “data are”.

Experimental design

Major comments:
1. The authors show multiple orientations and magnifications of the expression patterns, which is useful to validate their findings and serve as a reference for the community. However, I am confused about why the magnified images do not always correspond to wide-field. For instance, Fig 1A shows the entire animal and one purple patch of staining, but 1D shows the same animal at higher magnification with two sets of cranial ganglia. There are also differences in 1B and E, 3 A and C, and 5B and D. Why do they appear different? Are they different animals?

2. On a related note, the proportion of embryos that show each staining pattern is not clear. The authors say “The data is [are] representative of results of 2-3 rounds of in situ hybridization with at least 25 embryos per round”. However, do all embryos show the exact same pattern? Even with well-established probes, staining is often variable and dependent upon length of incubation with NBT/BCIP. The authors could express this as a fraction of animals with that pattern vs other patterns, no staining, nonspecific staining etc. In general, more discussion of variability would be helpful to understand how distinct each expression pattern is.

3. For many of the images, the microscopy contrast is appropriate to see anatomical structures, but others (particularly at 16.5 hpf) are harder to discern. Labeling additional landmark structures that were used to identify the regions with staining would be helpful (e.g. letter abbreviations for structure in addition to the arrows/arrowheads pointing out the staining.)

Minor comments:
1. I assume the wide-field and magnified images corresponds to 4x and 10x, as described in the methods. However, scale bars would be very useful in the figure itself.

2. The authors explain the reasoning for hydrolyzing the probes in the discussion (more could be in the methodology), but note in the methods that ntrk2a was not hydrolyzed. Was there a particular reasoning for this? Is there an expected difference in expression patterns?

Validity of the findings

Major concerns:
1. As noted above, the authors seem to have developed a set of novel probes, based on their reporting of primer sequences. However, the authors do not perform the standard control for whole-mount in situ hybridization, in which DIG-labeled sense probes are used alongside the antisense probes for comparison. Even though their staining does appear clear and as predicted in particular subsets of neurons, the sense control should be included to validate these new probes, both for the paper and for the community.

2. Alternatively, a functional assay demonstrating at least one of their predictions would really underscore the validity of their expression studies. While a CRISPR mutant would take a long time to develop, the authors could instead pharmacologically antagonize Trk receptors with acute treatment based on the window of expression. The authors note that Trk/p75 genes are necessary for neuron survival – do the expected neurons persist following receptor antagonism in zebrafish? This might address the cross-talk question nicely.

3. The authors state that in the discussion that their hydrolyzed, full-length probes would have the ability to detect multiple splice variants in an unbiased manner, unlike previous probes based on cDNA fragments. While I agree with that idea, I don't understand how this would produce a more conservative expression pattern, instead of broader to encompass different variants. Did the authors try a side-by-side with the other probes? Could it be differences in the protocol, NBT/BCIP timing, etc. rather than the probes?

Minor comment:
The authors describe potential signaling mechanisms based on the expression patterns and appropriately note the limitations of their study for functionality within the discussion. Though they summarize previous experiments in Table 2, they could experimentally compare their receptor staining patterns to ligand expression with co-staining. However, at minimum they need to cite the papers in the table itself.

Additional comments

Overall, this study has potential to be valuable to the zebrafish community. More controlled experiments should be done to validate specificity and variability of the expression probes and, ideally, show some functional implication for the expression patterns. Suggested text changes include discussion of other relevant literature and some re-working of text and figures.

Reviewer 3 ·

Basic reporting

Overall the manuscript is well written and clearly states what was previously known about these neurotrophin receptors. Figures likewise are well labelled with various arrows and arrowheads helping to orient the reader to the structures of interest.

There were a few minor typos I found.
In line 92 beginning should be used rather than begin
In line 215 there is an unnecessary and between 24 hpf and (Figure 5)
In the figure legend for Figure 1 there is a space breaking up “cranial”

Experimental design

As one of the key novel findings of this paper is the 16.5 hpf expression data I think it would be helpful to have a little bit more explanation in the introduction as to why this time point was chosen and what knowing expression at this time point adds to the big picture.

While overall I thought the methods were well written and provided a lot of useful details I felt there were a few things had could be added.

I did not see it mentioned in the methods how the animals were originally fixed. This information should be added.

While it is clear later in the manuscript it would be helpful to explicitly state in the probe synthesis section of the methods that these probes originated from full length cDNA.

The authors note that the ntrk2a probe was not hydrolyzed while other the others were. Clarification as to why this was would be helpful.

Validity of the findings

I felt overall the figures clearly showed the expression that was mentioned in the manuscript. However, there were a few instances that the stated expression was not clear from the included figures. Also while I really liked table 2 I think some additional information and formatting could make it more useful.

Based on the images provided it is not clear to me that ntrk3a is localized to the Rohan beard neurons. There is some purple in the tail area but it does not seem localized to the spinal cord but rather a haze throughout the tail. Is there a clearer image of this expression?

For ntrk3b it is mentioned in the discussion and table 2 that it is expressed in the lateral line primordium. However, this is not mentioned in the results or clearly shown in the figures. Are there figures that better show primordium expression?

Table 2 does not seem to include complete expression patterns for the genes. For example, Rohan beard neuron expression is not mentioned. Also for genes expressed in multiple tissue types it would be helpful to have it specifically noted which of those tissues the different ligands are expressed in.

Additional comments

Overall this was an interesting manuscript providing new insight into expression of neurotrophin receptors at an important developmental time point that had not previously been investigated.

---

## Round 0.2 · Minor Revisions

Thank you for resubmitting your manuscript to PeerJ with your thorough consideration of the three reviewers' comments. All three reviewers were very positive about the steps you took to address their concerns, including your additional validation of in situ probes, and have only minor remaining suggestion. Please consider these new reviewer comments with your next revision. Thank you as well for considering my suggested use of the public databases for zebrafish gene expression. I appreciate you looking into these resources and understand that they do not currently fit the scope of this manuscript.

I look forward to receiving your revision.

·

Basic reporting

The authors have nicely addressed comments and improve the manuscript.

Experimental design

No additional comments.

Validity of the findings

Although all reviewers raised several potentially interesting extensions or areas of investigation, the authors did a nice job focusing on the areas needed to solidify their results and interpretations. For example, the addition of controls and validations such as sense probes improves the manuscript. I have no additional comments.

Reviewer 2 ·

Basic reporting

The authors have improved the manuscript in this revision by including some key papers for comparison and improving their figures and tables for clarity.

There are some typos in the following lines:
120, missing period
145, capital A
159-160, line break
273, "ntr2a"

Experimental design

The sense probe data has greatly strengthened this manuscript, and I commend the authors for tackling the difficult though necessary experiments to validate their probes.

The new supplemental figure is well done, and I wonder if the authors could refer to it more, specifically in the results and even in the figure legends. For instance, in Figure 3, they say “expression is not detected” at 16.5 hpf when there is purple stain visible in the image (this nonspecific staining is what confused me the first time). Could they explicitly note non-specific staining with a reference to the supplemental figure? Given that a key takeaway is defining which genes are NOT expressed at 16.5 hpf, it would be good to make this clear. I am not sure it is as obvious to a reader what counts as "real" expression.

Validity of the findings

As noted above, the authors have heroically validated the specific expression observed. The points made in the discussion that relate to the data are more clear in this revised manuscript.

The one sense control that gives me pause is ntrk3b. The antisense and sense staining look similar in the new supplemental figure, and the antisense looks different from the image in Figure 6B. While not directly relevant to the takeaway message, it does relate to the bigger question about variability. The authors’ points about differences in staining being due to methodology, rather than variability in expression, are well taken. I agree that reporting fractions could be misleading. However, I still think some mention beyond "representative" is useful, especially if the microscope images are less clear than the investigators’ observations. It might save the PI time in the future fielding questions about these probes if the consistent vs. inconsistent staining and/or approximate (narrative, not numerical) frequency of background is noted somewhere in the manuscript, perhaps in a supplemental section to accompany the figure.

Reviewer 3 ·

Basic reporting

The revisions to the manuscript have addressed my concerns and put the findings in better context.

The only minor change I would like to see is a reference to the new Supplemental Figure 2 somewhere in the text of the manuscript. Perhaps it could be mentioned in the methods section that sense probes were examined to validate the specificity of the antisense probes.

Experimental design

My concerns have been addressed

Validity of the findings

My concerns have been addressed

---

## Round 0.3 · accepted · Accept

Thank you for resubmitting your manuscript and your work to address the remaining reviewer comments. All three original reviewers are now satisfied that you have addressed their suggestions and I am happy to now accept your paper for publication in PeerJ.

You will be given the option to make the reviews of your manuscript available to readers. Please consider doing so as this review record can be a great resource for readers of your paper and contributes to more transparent science.

Thank you for choosing PeerJ as a venue for publishing your work.